# Oxidation State and Local Structure of Chromium Ions in LaOCl

**DOI:** 10.3390/ma14133539

**Published:** 2021-06-25

**Authors:** Andris Antuzevics, Guna Krieke, Haralds Ozols, Andris Fedotovs, Anatolijs Sarakovskis, Alexei Kuzmin

**Affiliations:** Institute of Solid State Physics, University of Latvia, LV-1063 Riga, Latvia; guna.krieke@cfi.lu.lv (G.K.); haralds.ozols@cfi.lu.lv (H.O.); andris.fedotovs@cfi.lu.lv (A.F.); anatolijs.sarakovskis@cfi.lu.lv (A.S.); a.kuzmin@cfi.lu.lv (A.K.)

**Keywords:** LaOCl, chromium, X-ray photoelectron spectroscopy (XPS), X-ray absorption spectroscopy, electron paramagnetic resonance (EPR), electron-nuclear double resonance (ENDOR)

## Abstract

LaOCl doped with 0–10 mol% Cr was synthesized by thermal decomposition of chlorides. X-ray diffraction (XRD) analysis revealed that incorporation of chromium results in a decrease of the lattice parameter *a* and a simultaneous increase of the lattice parameter *c*. The local structure of chromium ions was studied with X-ray photoelectron (XPS), X-ray absorption (XANES), multifrequency electron paramagnetic resonance (EPR) and electron-nuclear double resonance (ENDOR) spectroscopy techniques. It was determined that synthesis in oxidizing atmosphere promotes the incorporation of chromium ions predominantly in the 5+ oxidation state. Changes of chromium oxidation state and local environment occur after a subsequent treatment in reducing atmosphere. Spin-Hamiltonian (SH) parameters for a Cr^5+^ and two types of Cr^3+^ centers in LaOCl were determined from the EPR spectra simulations.

## 1. Introduction

Lanthanum oxychloride (LaOCl), which is being considered for a variety of applications including gas sensors [1,2,3,4,5,6,7,8], catalysts [9,10,11], and solid electrolytes [12,13,14], is one of the most widely investigated compounds from the class of lanthanide oxyhalides. Historically, LaOCl has been primarily studied as a host matrix for rare-earth ions to establish the systematics of fluorescence spectra transition interrelations with crystal field parameters [15,16,17,18,19]. With appropriate activator combinations, it is an excellent phosphor, which can be tailored for efficient emission in spectral regions covering the whole visible [20,21,22,23,24,25,26,27,28,29,30,31,32,33,34,35,36,37,38,39,40,41] and a part of near-infrared regions [42,43].

LaOCl crystallizes in a tetragonal symmetry lattice with space group P4/nmm and cell parameters *a* = 4.12 Å and *c* = 6.88 Å [44,45]. As shown in Figure 1, the lattice consists of alternating cationic and anionic layers, which are arranged perpendicular to the *c* axis direction. La^3+^ ions occupy sites with C_4v_ symmetry and are coordinated by four O^2−^ ions from one side and five Cl^−^ ions from the other (with the fifth being in the next-to-nearest layer). Incorporation of trivalent rare-earth ions in LaOCl occurs via substitution of the single crystallographic La^3+^ position. Due to the relatively larger ionic radius of La^3+^, other trivalent rare-earth ions can be efficiently accommodated without distortions in cation site symmetry; however, a decrease in the unit cell parameters is expected [31,34,38,43]. The situation is more complicated for activators that have several stable oxidation states; for example, it is possible to stabilize both Eu^3+^ [27,28,29,30,31,32,33] and Eu^2+^ [33,34,35] ions in LaOCl during the synthesis. The larger Eu^2+^ ions expand the lattice and require charge compensation, possibly in the form of Cl^−^ ion vacancies [34]. The introduction of divalent cations, such as Ca^2+^ and Mg^2+^, is likely to produce lattice defects in LaOCl which can improve the optical [29] or electrical [13] properties of the material. Studies of Sb [46,47] and Bi [48] -doped LaOCl are especially interesting as the luminescence spectra showed the presence of two emission centers, while there is only one evident crystallographic site for substitution.

Despite the wide research performed on the host, the incorporation mechanisms of multi-valent impurities in the LaOCl lattice are not fully understood. Therefore, in this study we apply local structure and oxidation state sensitive X-ray and magnetic resonance spectroscopy techniques to elucidate the mechanism of multivalent chromium ion incorporation in LaOCl.

It is shown that during synthesis in oxidizing atmosphere, chromium ions enter into the structure of LaOCl predominantly in the 5+ state, producing asymmetric distortions of the lattice parameters. Local structure transformation of Cr upon annealing in reducing atmosphere is also discussed.

## 2. Experimental Details

LaOCl doped with 0–10 mol% Cr was prepared by thermal decomposition of chlorides. In a typical synthesis 3 mmol of LaOCl:Cr was prepared. An appropriate amount of CrCl_3_ 6H_2_O (99.5%) was dissolved in deionized water. Afterwards, La_2_O_3_ (99.999%) and 36.5% HCl (99.999%) were added. The obtained suspension was heated to boiling point until all LaOCl was dissolved and the excess water and HCl evaporated. The obtained material, consisting of chloride hydrates, was transferred to an Al_2_O_3_ crucible and heat-treated at 700 °C for 2 h in air. The heat treatment promoted the thermal decomposition of chlorides and the formation of LaOCl:Cr. Three parallel samples were prepared for all investigated dopant concentrations to test the reproducibility of the synthesis. To induce changes in the oxidation state of chromium ions, selected LaOCl:Cr samples doped with 0.1, 1 and 5% were additionally heat-treated in H_2_/Ar (5%/95%) at 500–800 °C for 2 h.

X-ray diffraction (XRD) patterns were obtained by Rigaku MiniFlex 600 diffractometer (Rigaku, Tokyo, Japan) with Bragg-Brentano θ-2θ geometry equipped with a 600 W Cu anode (Cu Kα radiation, λ = 1.5406 Å) X-ray tube operated at 40 kV and 15 mA. The Rietveld analysis was performed using the Profex software [50]. The Vesta code [49] was used for the visualization of the crystal structure.

XPS analyses were carried out using the ThermoFisher ESCALAB Xi^+^ instrument (Waltham, MA, USA) with a monochromatic Al Kα X-ray source. The instrument binding energy scale was calibrated to give binding energy at 932.6 eV for Cu 2p_3/2_ line of freshly etched metallic copper. For charge compensation, a standard procedure of sample surface irradiation with a flood of electrons was carried out. The spectra were recorded using an X-ray beam size of 900 × 10 microns, pass energy of 20 eV, and a step size of 0.1 eV. Data from all samples are referenced using the main signal of the carbon 1s spectrum assigned to occur at 284.8 eV. The carbon 1s spectrum was collected using high-energy resolution settings.

X-ray absorption spectroscopy study of chromium ion environment in LaOCl:Cr powder samples was performed at P65 Applied XAFS undulator beamline [51] of the PETRA III storage ring. A fixed exit double-crystal Si(111) monochromator was used to scan the energy range from 5300 eV to 6500 eV, and the harmonic rejection was achieved by an uncoated silicon plane mirror. The powder sample was attached to the sticky side of a Kapton tape, which was placed at 45° to the beam propagation direction. X-ray absorption spectra were collected at room temperature in fluorescence mode using an ionization chamber (*I*_0_) located before the sample and passivated implanted planar silicon (PIPS) detector (*I*_f_) placed at 90° to the incident beam. The X-ray absorption coefficient was calculated as *µ*(E) = *I*_f_/*I*_0_. X-ray absorption near edge structure (XANES) calculations at the Cr K-edge were performed using ab initio real-space FDMNES code [52,53] within the full-multiple-scattering (FMS) approximation. Only dipole transitions were taken into account, and the energy-dependent real Hedin-Lundqvist exchange-correlation potential was used [52,53]. The relativistic FMS calculations were performed with a self-consistent muffin-tin potential. The calculated XANES spectra were broadened to account for the core-hole level width of chromium (Γ(K-Cr) = 1 eV [54]) and other multi-electronic phenomena. The energy origin was set at the Fermi level *E*_F_. All calculations were performed for sufficiently large clusters with a radius of 8 Å around the absorbing chromium atom, which were constructed from the crystallographic LaOCl structure [45] by placing the chromium atom at the required place.

EPR spectra measurements were carried out at room temperature on Bruker ELEXSYS-II E500 CW-EPR system (Bruker Biospin, Rheinstetten, Germany) at X (9.83 GHz frequency; 10 mW power) and Q (33.85 GHz; 5.7 mW) microwave bands. Magnetic field modulation amplitude was 0.4 mT for measurements at both bands. Spectra intensities for the X-band EPR measurements have been normalized to sample mass. Electron nuclear double resonance (ENDOR) spectra were acquired with the same CW-EPR spectrometer equipped with DICE-II CW ENDOR measurement system and Bruker EN 901 X-Band CW-ENDOR resonator mounted on Oxford Instruments liquid helium flow cryostat. The temperature during data acquisition was 10 K, the magnetic field was 345.37 mT, and the microwave frequency was 9.5013 GHz at 20 mW power. The radiofrequency modulation type was FM with 100 kHz modulation depth. The resulting spectrum is a sum of 10 ENDOR scans. EPR and ENDOR spectra simulations were performed in EasySpin software [55].

## 3. Results and Discussion

Figure 2 shows XRD patterns of LaOCl samples doped with 0–10% Cr. The most intense diffraction peaks can be assigned to tetragonal LaOCl. Trace amounts of orthorhombic LaCrO_3_ could be detected in the samples doped with 5–10% Cr. Quantitative analysis of phase composition using Rietveld refinement revealed that LaCrO_3_ content is 4.7 ± 1.4% for the sample doped with 10% Cr and less than 4% in the rest of the samples. LaOCl samples with Cr concentrations below 5%, where impurity phase LaCrO_3_ was below the detection limit, were selected for further spectroscopic studies.

Gradual changes of lattice parameters with an increase of Cr content were detected (see Figure 3), suggesting the incorporation of Cr ions in the LaOCl lattice. In the case of isovalent substitution, Cr^3+^ ions are expected to incorporate in La^3+^ positions. Due to the smaller ionic radius of Cr^3+^ (0.615 Å) in comparison to La^3+^ (1.032 Å) [56], a contraction of the LaOCl lattice resulting in a decrease of the lattice parameters *a* and *c* is expected.

The calculated values of lattice parameters indicate a linear decrease of the lattice parameter *a* and a simultaneous increase of the lattice parameter *c*. This result suggests that the type of substitution is more complex than expected in the case of Cr^3+^ → La^3+^ substitution. LaOCl exhibits a layered structure, in which La-O layers alternate with Cl double layers. To prevent repulsion between Cl ions, the electron density is accumulated in the structural cavities between the Cl layers [45]. Similar anisotropic behavior has been reported for interstitial substitution in layered La_2_(Ni_0.9_M_0.1_)O_4+δ_ (M = Fe, Co, and Cu) perovskite [57] and metal alloy [58,59] materials. We propose that in the case of the introduction of chromium in LaOCl, Cr ions might incorporate between the Cl layers, thus increasing the lattice parameter *c*, or be located in an off-site position in the case of La substitution.

Results of the X-band measurements for LaOCl samples with different concentrations of Cr are presented in Figure 4. No EPR signals could be detected in the undoped sample. EPR spectra of chromium doped samples consist of a broad line at 358 mT (*g* ≈ 1.96). There is an obvious correlation between its intensity and activator concentration, which is a strong indication that the signal belongs to a Cr-related paramagnetic center.

EPR signals associated with Cr^5+^ and Cr^3+^ oxidation states have been reported to be detectable at room temperature; for the measurements of transition metal ions with an even number of electrons, low temperatures are usually required [60]. The more commonly encountered form is Cr^3+^, which is a spin *S* = 3/2 system. In a local cubic symmetry, the EPR spectrum consists of a single central line in *g* ≈ 1.97–1.98 range with a characteristic ^53^Cr hyperfine (HF) structure [61,62,63]. In the case of lower local symmetry, several transitions in a field range determined by the magnitude of zero-field splitting (ZFS) of the ground state can be expected [63,64,65,66,67,68]. Cr^5+^ is a *S* = 1/2 system; therefore, there are no effects related to ZFS, and the EPR signal is not as complicated as for Cr^3+^. A notable feature occasionally observed in the spectra of Cr^5+^ is HF structure caused by an interaction with the nearest cations [69,70,71,72]. We can observe a similar effect for the 2 mol% sample; however, due to the high Cr concentration, it is partially obscured. A sample with 0.1 mol% Cr was therefore synthesized for a more detailed analysis of chromium incorporation in LaOCl; the results of combined EPR and ENDOR investigations are shown in Figure 5.

The EPR spectrum exhibits a well-resolved HF structure with approximately 1.15 mT average separation between the lines, which corresponds to a HF coupling value of *A* ≈ 32 MHz. The number and relative intensities of the lines could be tentatively accounted for in a model with *S* = 1/2 interacting with four equivalent nuclear spins *I* = 7/2. From the naturally abundant nuclei in the LaOCl matrix, only ^139^La have *I* = 7/2. ENDOR spectra, in general, are determined by the relative magnitude of HF and nuclear Zeeman interactions [73]. The fact that we observe the ENDOR signal at *A*/2 ≈ 16 MHz implies that our case corresponds to a Larmor-split HF-centered doublet. Consequently, the nucleus responsible for the EPR spectrum HF structure can be identified from half the distance between the ENDOR lines. As a result, we can estimate that the Larmor frequency is 2.1 MHz directly from the experiment, which is close to the value of ^139^La at *B* = 345 mT (the magnetic field at which ENDOR was acquired). The asymmetric shape of the ENDOR signal suggests that the HF interaction is anisotropic. A spin-Hamiltonian (SH) with axial HF interaction was used for EPR and ENDOR spectra simulations:(1)$$\hat{H}=g\mu_{B}B\cdot \hat{S}+\hat{S}\cdot A\cdot \hat{I}$$
where g is the *g*-factor; $\mu_{B}$—the Bohr magneton; B—external magnetic field; A—the HF coupling tensor with components $A_{┴}=A_{x}=A_{y}$ and $A_{\|}=A_{z}$ [74]. The best simultaneous fit to the experimental EPR and ENDOR spectra was achieved with the following SH parameter values: *g* = 1.964 ± 0.001, $A_{┴}$ = 30.72 ± 0.10 MHz, and $A_{\|}$ = 37.63 ± 0.10 MHz. The lineshape of the EPR signal was modelled with a Gaussian distribution of energy levels (EasySpin HStrain function [55]). The determined full widths at half maximum (FWHM) of the distributions for perpendicular and parallel directions were 33 and 50 MHz respectively. The linewidth used for the ENDOR simulation was 1 MHz. The main result from magnetic resonance spectroscopy analysis can be summarized as follows: the initial incorporation of chromium occurs in the Cr^5+^ state in a site, which is coordinated by 4 equivalent La^3+^ ions.

Annealing in reducing (H_2_/Ar) atmosphere was carried out to induce transformations of the oxidation state of chromium ions. No changes in the phase composition of LaOCl:Cr were detected after the heat treatment at 500–800 °C for 2 h. The results of XPS analysis of chromium 2p peaks for the samples annealed at different temperatures are shown in Figure 6.

The XPS spectrum contains strong signals of La, O, and Cl; however, Cr signal intensity is miniscule due to the relatively low doping content. Nevertheless, two peaks can be resolved in the Cr 2p transition range, which correspond to Cr 2p_3/2_ and Cr 2p_1/2_ spin-orbit splitting doublets. The aforementioned peaks monotonously shift toward smaller energies, when the annealing temperature is increased. Complex multiplet splitting of Cr peaks with up to five components has been demonstrated for different compounds [75]. To establish the effect of the annealing temperature on the XPS signal of chromium, the binding energy of the 2p_3/2_ peak was calculated as a sum of weighed binding energies of the subpeaks for every annealing temperature. The obtained dependency is shown in Figure 6b. A summary of binding energy values for the 2p_3/2_ peak in chromium oxides is given in Table 1.

The experimentally measured binding energy values for the peak 2p_3/2_ for different annealing temperatures fall in the set of values characteristic for Cr^n+^, where *n* varies from *n* = 5 for the annealing temperature of 400 °C to *n* = 3 for the annealing temperature of 800 °C.

Experimental XANES spectra recorded in the range of the La L_1,2,3_-edges and Cr K-edge for 2% Cr LaOCl samples before and after heat-treatment in H_2_/Ar at 800 °C for 2 h are shown in Figure 7. As one can see, the Cr K-edge is located between the La L_1_ and L_2_ edges that complicates its analysis. Moreover, the XANES spectra are distorted by the presence of several “glitches” (Bragg reflections from the crystal monochromator), which occur as vertical spikes. Glitches can be uniquely identified by inspecting the I_0_ signal measured by the ionization chamber located before the sample. Three of them (one at 5985.8 eV and two at 5998.8 and 6000.1 eV) are, unfortunately, located at the beginning of the Cr K-edge (Figure 7b). Nevertheless, there is a notable difference between the two samples. The as-prepared sample contains a pre-edge peak at 5992.4 eV, which is completely absent in the reduced sample. Besides this difference, there are also other changes in the fine structure above the edge, which distinguish the two samples: in particular, the peak located at 6015 eV is broadened in the reduced sample. The pre-edge peak is attributed to the 1s → 3d(Cr) transition, which is forbidden in a dipole approximation for geometries that possess an inversion center but becomes allowed in a non-centrosymmetric environment, e.g., tetrahedral [81,82]. This means that the treatment in reducing atmosphere strongly affects the local environment of chromium ions.

Both Cr^3+^ (0.62 Å) and Cr^5+^ (0.35–0.57 Å) have smaller ionic radii than La^3+^ (1.16–1.22 Å) [56]; therefore, a substitution of lanthanum ions by chromium ions is structurally possible. Moreover, crystalline CrOCl_3_ is known [83], where chromium ions are present in the oxidation state 5+ and have a local environment, which is partially close to that of La in LaOCl. In CrOCl_3_ chromium ions have square pyramidal coordination with four chlorine atoms forming a base of the pyramid with *R*(Cr-Cl) = 2.16–2.34 Å and one oxygen atom located at the pyramid top with *R*(Cr-O) = 1.54 Å [83]. The small size of Cr^5+^ ions also allows us to suggest an alternative placement in LaOCl structure voids present within the layers of chlorine ions. In this case, chromium ions will have tetrahedral coordination by four chlorine atoms.

To discriminate between different structural models, the Cr K-edge XANES spectra were calculated as described above and are shown in Figure 8. In the case of lanthanum substitution (Figure 8a), the influence of chromium atom displacement along the *c*-axis by Δz = −0.20… + 0.15 Å has been additionally evaluated. The obtained results suggest that the pre-edge peak observed in the experimental XANES for the as-prepared sample can be reproduced using both structural models, i.e., a non-centrosymmetric environment of Cr^5+^ ions can be associated with their location in tetrahedral [CrCl_4_] (Figure 8b) or distorted [CrO_4_Cl_4_] (Figure 8a) surroundings. In the latter case, the chromium ions should be displaced closer to the oxygen ions by about 0.1 Å, which is favored by the smaller size of Cr^5+^ ions and stronger Cr^5+^-O^2−^ Coulomb interaction. Upon reduction to the oxidation state 3+ in the reduced sample, the size of the chromium ions increases and the interaction with oxygen ions becomes weaker, which promotes the displacement of chromium ions toward lanthanum position or even slightly above it. Such displacement leads to a drastic reduction of the pre-edge peak intensity and a lowering of the amplitude of the peak located at 15–20 eV above the Fermi level in Figure 8a (it corresponds to the peak at 6015 eV in the experimental XANES in Figure 7b). Thus, the substitutional model, in which chromium ions are located close to the lanthanum position, can describe the experimentally observed variations in the Cr K-edge XANES in the as-prepared and reduced LaOCl:Cr samples.

X-band EPR spectra of 1% Cr samples annealed at different temperatures in the H_2_/Ar atmosphere are shown in Figure 9.

A gradual decrease of the Cr^5+^ signal intensity is observed as the reducing temperature is increased, which is accompanied by an emergence of additional signals over a broader field range. The number of features in the spectrum and the fact that EPR was detected at room temperature suggest the reduction of Cr^5+^ to Cr^3+^. EPR measurements at two frequency bands were performed for a more detailed analysis of the Cr^3+^ local structure; the results are shown in Figure 10.

One of the advantages of measurements at higher microwave frequencies is a simplification of spectra for *S* > 1/2 systems with a large magnitude of ZFS [84]. Moreover, interpretation of experimental results is more unambiguous, if a simultaneous fit for EPR spectra acquired at different frequencies can be achieved with the same parameter set. Simulations were performed with the following SH:(2)$$\hat{H}=g\mu_{B}B\cdot \hat{S}+\hat{S}\cdot D\cdot \hat{S}$$
where D is the ZFS tensor, which can be reduced to two parameters: *D* and *E* [85] and *S* = 3/2. The determined SH parameter values are summarized in Table 2.

The final simulation curves were achieved as a superposition of the initial Cr^5+^ signal and two Cr^3+^ centers labelled as I and II respectively. Both Cr^3+^ centers are axially symmetric (rhombic ZFS parameter *E* = 0); however, the magnitude of ZFS differs by a factor of two indicating significant variations in the local environment. The relative contribution (EasySpin Weight function) from the Cr^3+^ I center was determined to be higher by a factor of 30, thereby implying that this site is dominant in LaOCl. There was a feature (530 mT at X-band and 1135 mT at Q-band) in the experimental spectra, which could not be accounted for in the simulations. Its most likely origin is associated with another type of Cr-related defect.

The results of X-ray and magnetic resonance spectroscopy analysis demonstrate that oxidation state and local structure of chromium ions in LaOCl can be controlled during synthesis. The solubility limit of Cr in LaOCl is approximately 5 mol%; at higher concentrations formation of LaCrO_3_ impurity phase can be detected. In the investigated Cr concentration range, all XRD patterns can be ascribed to the tetragonal LaOCl phase without detectable deviations in crystal symmetry. The incorporation of Cr ions in the crystal structure can be deduced from the shift of XRD peak positions. Rietveld analysis reveals an asymmetric distortion of the lattice, that is, a decrease of the lattice parameter *a* and simultaneous increase of the lattice parameter *c*. The relatively small level of Cr doping prevents a direct detection of Cr local environment in LaOCl host matrix by XRD; therefore, local structure sensitive tools were used to probe the local environment around Cr ions in this study. As evidenced by the EPR and XPS data, if synthesized in air, the incorporation of chromium in LaOCl occurs predominantly in the 5+ oxidation state. The EPR and ENDOR spectra of LaOCl:Cr^5+^ can be simulated in a *S* = 1/2 model with HF interaction with four equivalent La nuclei. XANES spectra of the corresponding samples contain the forbidden 1s→3d(Cr) pre-edge structure, which is an indication of a non-centrosymmetric environment around Cr^5+^ ions. The observation can be explained by two different structural models: Cr^5+^ placement in the voids within the layers of chlorine ions or in an off-site position in the case of La substitution (in both models Cr^5+^ HF interaction with 4 equivalent La nuclei could be expected).

Annealing in a reducing atmosphere produces changes in the oxidation state and local environment of chromium ions. A gradual Cr^5+^→Cr^3+^ transformation is observed in the XPS spectra as the reducing temperature increases. EPR spectra simulations reveal that at least two Cr^3+^ centers are formed in the LaOCl structure as the result. However, a possibility of other oxidation states cannot be excluded (EPR signal of chromium having oxidation states with an even number of electrons is not expected to be detectable at our experimental conditions). It is likely that the dominant contribution to the EPR spectrum originates from Cr^3+^ substituting La^3+^ ions. Such geometry possesses an inversion center, which explains the absence of XANES 1s→3d(Cr) pre-edge structure in the spectrum of the reduced sample. The obtained results show that accommodation of small cations produces distortions to the LaOCl structure, which could be a prospective strategy for tailoring optical, electrical, or catalytical properties of the material.

## 4. Conclusions

The oxidation state and local environment of chromium ions were studied in LaOCl samples synthesized in different annealing atmospheres. A strong effect of chromium content on the LaOCl lattice was evidenced by X-ray diffraction. XPS, XANES, multifrequency EPR and ENDOR spectroscopy techniques were used to elucidate how chromium ions are embedded into the matrix.

Surprisingly, we found that chromium ion incorporation in LaOCl occurs predominantly in the 5+ oxidation state, producing asymmetric distortions to the crystal structure: a decrease of the lattice parameter *a* and an increase of the lattice parameter *c*. The EPR spectrum of LaOCl:Cr^5+^ consists of a resonance at *g* = 1.964 with a well-resolved HF structure from interaction with four equivalent La nuclei. XRD and EPR data, which are validated by XANES calculations, suggest that Cr^5+^ incorporation occurs either in the voids within the layers of chlorine ions or in an off-site position in the case of La substitution. Annealing in a reducing atmosphere affects the oxidation state and local structure of chromium ions so that at least two types of axial symmetry Cr^3+^ centers are formed in LaOCl.

## Figures and Tables

**Figure 1 materials-14-03539-f001:**
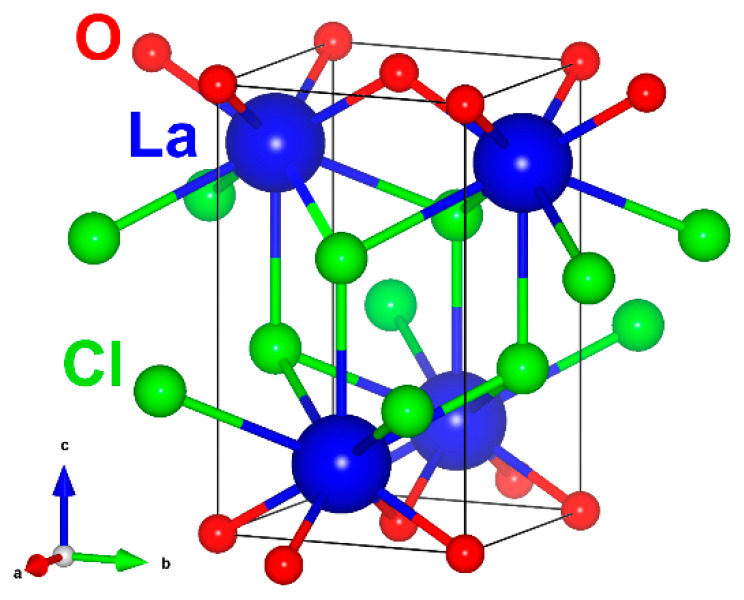
Crystal structure of LaOCl. Visualized in VESTA [49] using atom coordinates from [45].

**Figure 2 materials-14-03539-f002:**
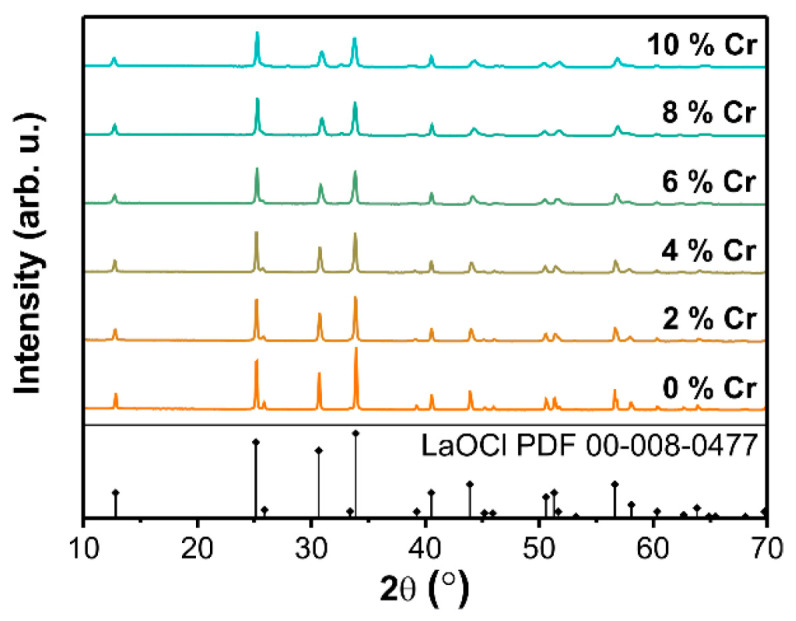
XRD patterns of LaOCl samples with different concentrations of Cr.

**Figure 3 materials-14-03539-f003:**
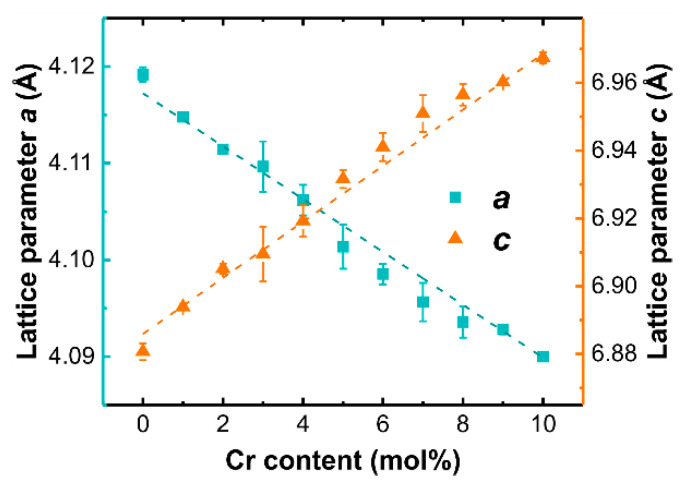
LaOCl lattice parameter dependence on the level of Cr content.

**Figure 4 materials-14-03539-f004:**
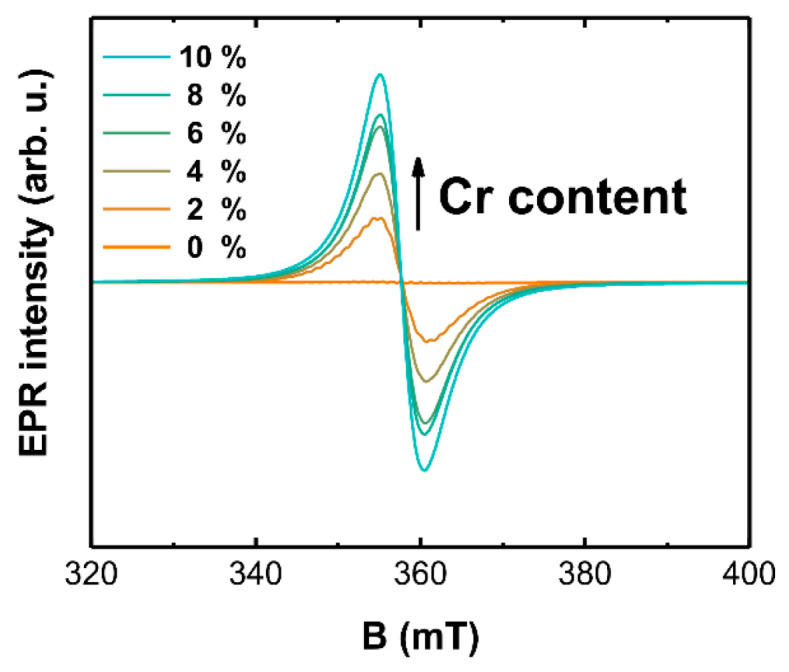
EPR spectra of LaOCl samples with different concentrations of Cr.

**Figure 5 materials-14-03539-f005:**
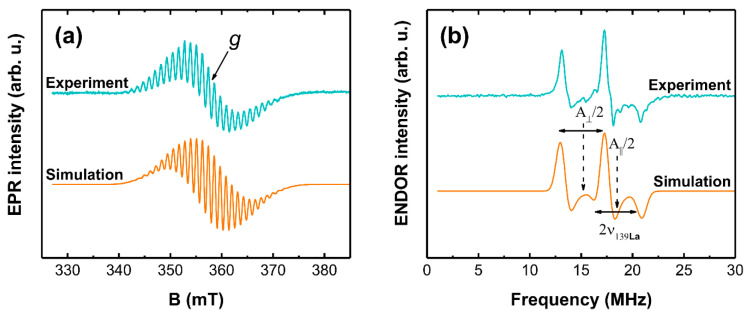
(**a**) EPR and (**b**) ENDOR spectra simulations of LaOCl sample with 0.1 mol% Cr.

**Figure 6 materials-14-03539-f006:**
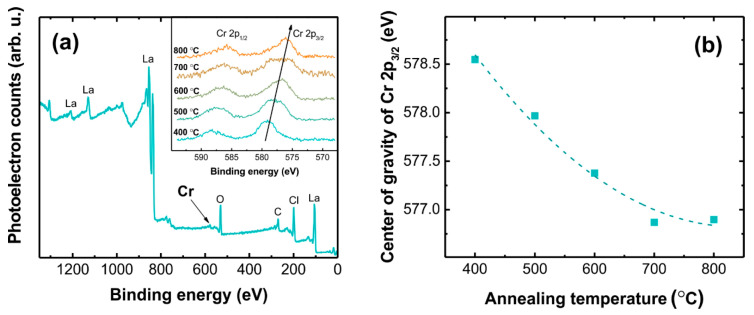
(**a**) XPS spectrum of the 5% Cr LaOCl sample prior annealing in a reducing atmosphere; inset: background corrected XPS spectra in Cr 2p peak range of 5% Cr LaOCl samples annealed at different temperatures in H_2_/Ar atmosphere and (**b**) binding energy of Cr 2p_3/2_ to the annealing temperature of the samples.

**Figure 7 materials-14-03539-f007:**
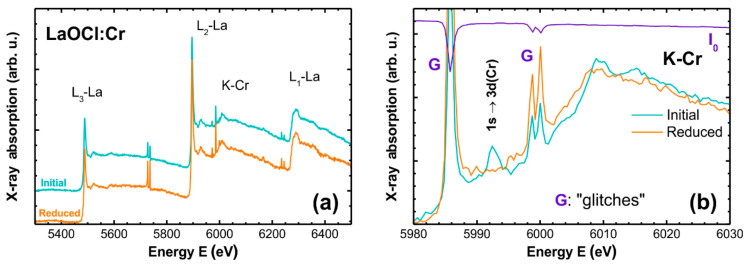
Experimental XANES spectra recorded in the range of the La L_1,2,3_-edges and Cr K-edge for 2% Cr LaOCl samples before and after annealing in reducing atmosphere. The I_0_ signal measured by the ionization chamber located before the sample is shown at the top in (**a**). Several “glitches” are observed and labelled with the letter “G” in (**b**).

**Figure 8 materials-14-03539-f008:**
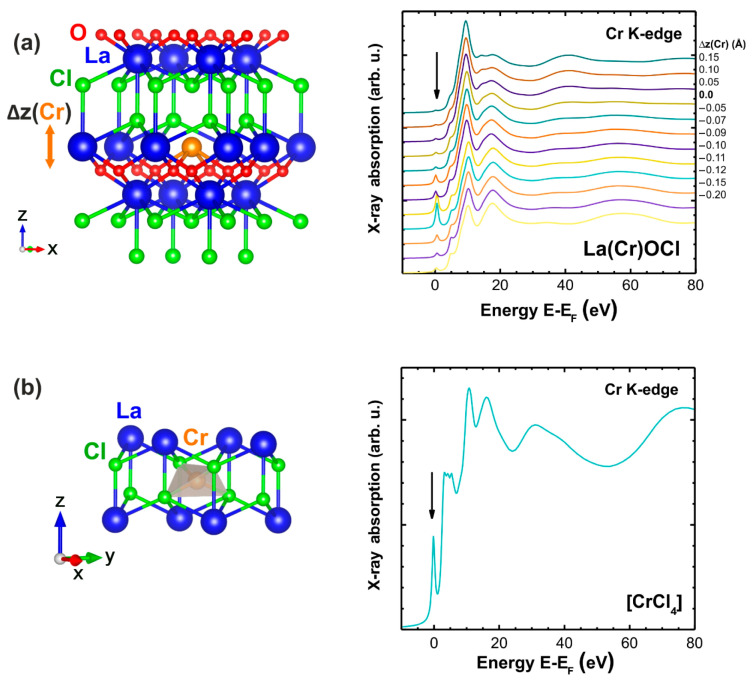
(**a**) A structural model of the lanthanum ion substitution by chromium (left). Calculated Cr K-edge XANES spectra for different positions of chromium atoms displaced along the *c*-axis by Δz relative to the lanthanum position (right). (**b**) A fragment of the structural model of the chromium environment with the tetrahedral coordination by chlorine atoms (left). The calculated Cr K-edge XANES spectrum (right). The pre-edge peak is indicated by arrows.

**Figure 9 materials-14-03539-f009:**
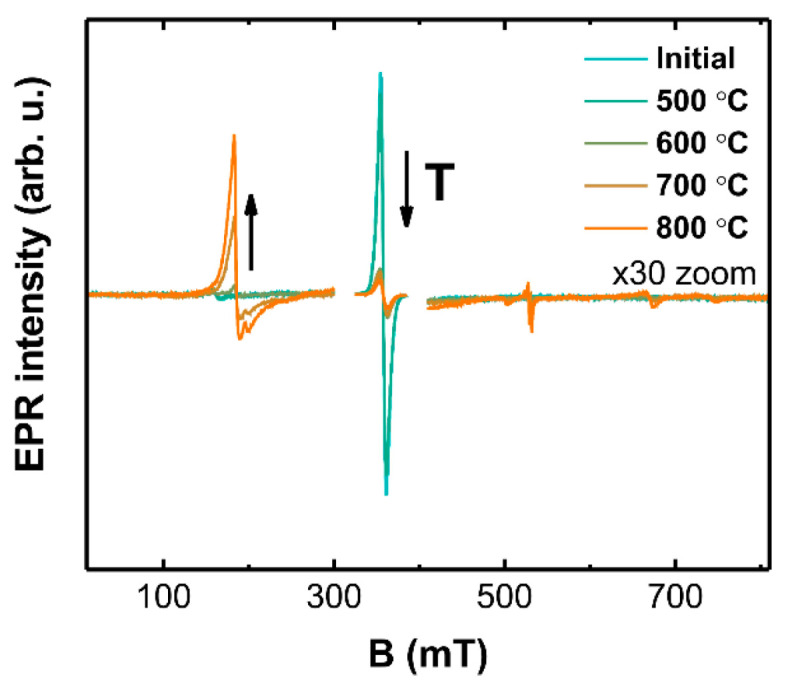
EPR spectra of 1% Cr LaOCl samples after annealing at different temperatures in reducing atmosphere. Spectra intensities in 0–300 and 400–800 mT ranges have been magnified 30-fold.

**Figure 10 materials-14-03539-f010:**
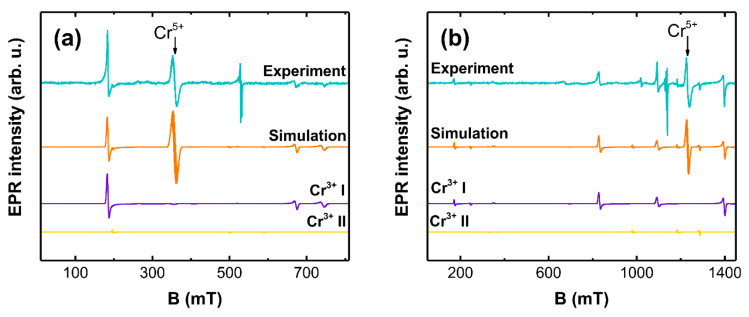
(**a**) X-band and (**b**) Q-band EPR spectra simulations of 0.1% Cr LaOCl sample after annealing at 800 °C in reducing atmosphere.

**Table 1 materials-14-03539-t001:** A summary of 2p_3/2_ binding energies in chromium oxides.

Binding Energy, eV	Compound	Reference
576.7–577.2	Cr_2_O_3_	[75]
577.1	Cr_2_O_3_	[76]
577.0	Cr_2_O_3_	[77]
577.0	CrO_2_	[78]
576.6	CrO_2_	[79]
579.3	Cr_2_O_5_	[76]
579.0	Cr_2_O_5_	[77]
579.3	CrO_3_	[78]
580.3	CrO_3_	[80]
580.0	CrO_3_	[77]
578.5	Annealed at 400 °C	Current work
576.9	Annealed at 800 °C	Current work

**Table 2 materials-14-03539-t002:** Fitted SH parameter values of Cr^3+^ centers in LaOCl.

Cr^3+^ Center	*g*	*D*, MHz	HStrain, MHz
I	1.973 ± 0.001	12063 ± 200	253 ± 25
II	1.977 ± 0.001	6865 ± 100	115 ± 15

## Data Availability

The data presented in this study are available in article.

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
