# Peer review of "Oxidation State and Local Structure of Chromium Ions in LaOCl"

_materials, 2021, doi:10.3390/ma14133539_

Round 1
Reviewer 1 Report
This paper presents a multifaceted analysis of the chemical state and local structure of Cr in LaOCl, which is expected to be used for various applications, when La is replaced by Cr. The experimental results obtained in the present work are interesting and the interpretation of the experimental and simulated data is generally well described. In this sense, the paper is considered worthy of publication in this journal. However, there are some points that need to be added or discussed, and the reviewer would like to ask for the authors’ consideration.
- The authors report that the lattice parameter of the a-axis (c-axis) increases (decreases) with Cr doping through X-ray diffraction experiments; if the local structures around La or Cr are in variant with respect to Cr doping, it is unlikely to be obvious that the space group of the crystal structure is invariant with Cr doping. Since the Cr doping dependence of the crystal structure affects the discussion of the results of XAFS described below, the reviewer would like to request a detailed description of the Cr doping dependence of the crystal structure.
- Is the observable correct for XANES or X-ray absorption, in the XANES spectrum? In addition, the spectrum shown in Fig. 7 seems to include not only XANES but also XAFS. Although detailed analyses are difficult due to the proximity of multiple absorption edges, XAFS spectra reflect the local structure. It is also necessary to discuss the comparison of the results of XAFS with those of the X-ray diffraction in the present work.
- The sample treatment is different for the EPS experiment from the samples used for the other experiments. To help the readers understand, it is necessary to show the reasons for the different sample treatment. Also, in the discussion, only the possible chemical states of Cr3+ and Cr5+ are discussed. A more detailed discussion is needed on the specific Cr chemical states in LaOCl.
- As can be mentioned throughout, this paper does not describe any relationship between the results of all the experiments, even though multiple experiments are conducted. It is essential to discuss the results of the all the experiments that have been conducted in the present work.
Adding clear descriptions shown above are necessary for this paper to be accepted for publication in this journal.
Author Response
Please find the response attached.

Reviewer 2 Report
The manuscript is well written and the experimental work seems to have been done with scientific criteria.
From my point of view, the manuscript can be published without corrections.
Author Response
Please find the response attached.

Reviewer 3 Report
Present article is rather good written and can be interested for the scientists working in inorganic chemistry, materials science and solid state physics.
Nevertheless some points must be improved or explained.
The data are presented very clear, but all of my questions originated from the fact, that the subsituted phases under discussion contain up to 6% of impurity of LaCrO3 determined by XRD, which is known as not very precise analytical technique.
Are there any ways to make synthesis better to reach equlibrium?
When synthesis repeated will the impurity content be same? Do authors investigated the phase diagram of the system?
How the impuriry can influence on XANES, EPR and XPS results?
Why whole XPS spectrum is not provided?
While the ionic size of La and Cr differ in more than 2 times why the c and especially a parameters change unsufficient?
Did authors measured the ionic conductivity, how the Cr substitution can influence on it?
Author Response
Please find the response attached.

Round 2
Reviewer 1 Report
The manuscript sent by the authors still needs to be revised, although the main text has been revised in response to most of the comments. The reviewer requests the authors to revise the manuscript as follows: The authors misunderstood the “observable” in XANES and EXAFS measurements. The definition of the vertical axes in Fig. 8(a) and (b) seems to be wrong. The referee believes that the observable in both XANES and EXAFS measurements is X-ray absorption in arbitrary units experimentally as Fig. 7 in this paper. If the textbooks on XAFS define the energy region where the XANES signals come, the authors should cite these textbooks or papers as references. Otherwise, the vertical axis of Fig. 8 should be X-ray absorption instead of XANES in the referee’s best knowledge. After the revision based on the present comment, the reviewer recommends to the editor that this manuscript is worth to be published in this journal.
